# Associations between Normal Organs and Tumor Burden in Patients Imaged with Fibroblast Activation Protein Inhibitor-Directed Positron Emission Tomography

**DOI:** 10.3390/cancers14112609

**Published:** 2022-05-25

**Authors:** Aleksander Kosmala, Sebastian E. Serfling, Niklas Dreher, Thomas Lindner, Andreas Schirbel, Constantin Lapa, Takahiro Higuchi, Andreas K. Buck, Alexander Weich, Rudolf A. Werner

**Affiliations:** 1Department of Nuclear Medicine, University Hospital Würzburg, 97080 Würzburg, Germany; serfling_s1@ukw.de (S.E.S.); dreher_n@ukw.de (N.D.); lindner_t3@ukw.de (T.L.); schirbel_a@ukw.de (A.S.); thiguchi@me.com (T.H.); buck_a@ukw.de (A.K.B.); werner_r1@ukw.de (R.A.W.); 2Nuclear Medicine, Faculty of Medicine, University of Augsburg, 86156 Augsburg, Germany; 3Faculty of Medicine, Dentistry and Pharmaceutical Sciences, Okayama University, Okayama 700-8530, Japan; 4Gastroenterology, Department of Internal Medicine II, University Hospital Würzburg, 97080 Würzburg, Germany; weich_a@ukw.de; 5The Russell H Morgan Department of Radiology and Radiological Sciences, Johns Hopkins School of Medicine, Baltimore, MD 21205, USA

**Keywords:** PET, [^68^Ga]Ga-FAPI, theranostics, radioligand therapy, fibroblast activation protein

## Abstract

**Simple Summary:**

Several radiolabeled fibroblast activation protein targeted inhibitors (FAPI) have been developed for molecular imaging and therapy. A potential correlation of radiotracer uptake in normal organs and extent of tumor burden may have consequences for a theranostic approach using ligands structurally associated with [^68^Ga]Ga-FAPI, as one may anticipate decreased doses to normal organs in patients with extensive tumor load. In the present proof-of-concept study investigating patients with solid tumors, we aimed to quantitatively determine the normal organ biodistribution of [^68^Ga]Ga-FAPI-04, depending on the extent of tumor. Except for a trend towards significance in the myocardium, we did not observe any relevant associations between PET-based tumor burden and normal organs. Those preliminary findings may trigger future studies to determine possible implications for theranostic approaches and FAP-directed drugs, as one may expect an unchanged dose for normal organs even in patients with higher tumor load.

**Abstract:**

(1) Background: We aimed to quantitatively investigate [^68^Ga]Ga-FAPI-04 uptake in normal organs and to assess a relationship with the extent of FAPI-avid tumor burden. (2) Methods: In this single-center retrospective analysis, thirty-four patients with solid cancers underwent a total of 40 [^68^Ga]Ga-FAPI-04 PET/CT scans. Mean standardized uptake values (SUV_mean_) for normal organs were established by placing volumes of interest (VOIs) in the heart, liver, spleen, pancreas, kidneys, and bone marrow. Total tumor burden was determined by manual segmentation of tumor lesions with increased uptake. For tumor burden, quantitative assessment included maximum SUV (SUV_max_), tumor volume (TV), and fractional tumor activity (FTA = TV × SUV_mean_). Associations between uptake in normal organs and tumor burden were investigated by applying Spearman’s rank correlation coefficient. (3) Results: Median SUV_mean_ values were 2.15 in the pancreas (range, 1.05–9.91), 1.42 in the right (range, 0.57–3.06) and 1.41 in the left kidney (range, 0.73–2.97), 1.2 in the heart (range, 0.46–2.59), 0.86 in the spleen (range, 0.55–1.58), 0.65 in the liver (range, 0.31–2.11), and 0.57 in the bone marrow (range, 0.26–0.94). We observed a trend towards significance for uptake in the myocardium and tumor-derived SUV_max_ (ρ = 0.29, *p* = 0.07) and TV (ρ = −0.30, *p* = 0.06). No significant correlation was achieved for any of the other organs: SUV_max_ (ρ ≤ 0.1, *p* ≥ 0.42), TV (ρ ≤ 0.11, *p* ≥ 0.43), and FTA (ρ ≤ 0.14, *p* ≥ 0.38). In a sub-analysis exclusively investigating patients with high tumor burden, significant correlations of myocardial uptake with tumor SUV_max_ (ρ = 0.44; *p* = 0.03) and tumor-derived FTA with liver uptake (ρ = 0.47; *p* = 0.02) were recorded. (4) Conclusions: In this proof-of-concept study, quantification of [^68^Ga]Ga-FAPI-04 PET showed no significant correlation between normal organs and tumor burden, except for a trend in the myocardium. Those preliminary findings may trigger future studies to determine possible implications for treatment with radioactive FAP-targeted drugs, as higher tumor load or uptake may not lead to decreased doses in the majority of normal organs.

## 1. Introduction

Fibroblast activation protein (FAP) is a type II membrane-bound glyocoprotein acting as serin protease of the dipeptidyl-peptidase family and can be detected on the cell surface of cancer-associated fibroblasts, which have been advocated to play a relevant tumor-promoting role [1,2,3,4,5,6]. Moreover, FAP is also overexpressed by stromal fibroblasts in over 90% of epithelial carcinomas [2,7]. Several radiolabeled FAP-targeted inhibitors (FAPI) have been developed for FAP-directed molecular imaging [8,9]. Furthermore, FAP may also be targeted in a theranostic approach, e.g., by labelling with ß-emitters such as ^177^Lu or ^90^Y [8,10,11,12]. A potential correlation of radiotracer uptake in normal organs and extent of tumor burden, however, may have consequences for using such therapy ligands structurally associated with [^68^Ga]Ga-FAPI, as one may anticipate decreased doses to normal organs in patients with extensive tumor load. Other applications include “cold” FAP-directed drugs, as relevant off-target effects may be reduced in individuals with high tumor load [13]. While some studies have shown that other PET agents also used in a theranostic setting may be susceptible to such effects of tumor burden on radiotracer uptake in normal organs [14,15,16,17], other studies have demonstrated either no such effect, or only minimal impact of PET-avid tumor load [18,19,20]. Thus, results from previous studies cannot be extrapolated to FAP-directed molecular imaging.

In the present proof-of-concept study, we aimed to quantitatively investigate the normal organ biodistribution of [^68^Ga]Ga-FAPI-04 in patients with solid tumors, depending on the extent of tumor burden and to establish potential correlations between radiotracer accumulation in normal organs and uptake in FAPI-avid disease sites. Additionally, as increased FAPI uptake in scar tissue after previous surgical procedures may be an important confounding variable [21], we investigated potential correlations between normal organ radiotracer accumulation and FAPI uptake in surgical routes.

## 2. Materials and Methods

In this retrospective study, we analyzed 34 patients with known solid malignant tumors, which received 40 [^68^Ga]Ga-FAPI-04 PET/CT scans at our institution. All patients gave their written informed consent to undergo [^68^Ga]Ga-FAPI-04 PET/CT imaging. The local institutional review board waived the need for further approval due to the retrospective character of this study (No. 20210415 02). Parts of this cohort have been investigated previously [22,23,24,25], without assessing correlations between normal uptake and tumor burden.

### 2.1. Imaging Procedure and Analysis

Synthesis and radiolabeling of [^68^Ga]Ga-FAPI-04 was performed as described previously [8]. [^68^Ga]Ga-FAPI-04 PET/CT scans were conducted using a Siemens Biograph mCT 64 or 128 (Siemens Healthineers, Erlangen, Germany). Approximately 60 min after injecting of 148 MBq (median; range, 79–239 MBq) whole body scans (range: vertex-proximal thighs) were acquired. PET scans were performed in 3D mode with an acquisition time of 3–5 min/bed position, as described in [25]. PET images were analyzed using syngo.via software (version VB60A; Siemens Healthineers, Erlangen, Germany). We assessed PET, CT, and hybrid PET/CT overlay for every scan. [^68^Ga]Ga-FAPI-04 biodistribution in normal organs was determined by placing respective volumes of interest (VOIs) as described in [9,18,19,20]. This approach allowed us to include a total of 10 VOIs per patient in the liver, spleen, pancreas, left and right kidney, the left myocardial septal wall and lateral wall, and in the vertebral bodies of C2, Th7 and L5. To evaluate radiotracer uptake in the heart and in the bone marrow, the average of the respective VOIs was used. Mean standardized uptake values (SUV_mean_) were then recorded for normal organs [18,19,20]. Total tumor burden was also quantified by drawing spherical volumes of interests. The software automatically adapted a three-dimensional VOI at a 40% isocontour and averaged maximum standardized uptake values (SUV_max_; averaged over all tumor lesions per scan), sum of tumor volume (TV, in cm^3^), and sum of fractional tumor activity (FTA = TV × SUV_mean_) were recorded [18,19,20]. Tracer uptake in post-surgical areas was quantified in a similar manner, and SUV_max_ (averaged over all areas showing increased post-surgical uptake per scan), sum of post-surgical FAP-uptake volume (PS-FV in cm^3^ per scan), and sum of post-surgical fractional FAP-activity (PS − FFA = PS − FV × SUV_mean_; per scan) were recorded. To reduce partial volume effects, lesions smaller than 15 mm or 1.7 cm^3^ were not included [26]. We differentiated tumor lesions in the following organ compartments: primary tumor, liver, lung, skeleton, lymph nodes, and soft tissue. The potential impact of tumor burden on normal organ radiotracer uptake would be expected to be more pronounced at higher tumor burden levels. Thus, we removed patients with a sum of total tumor volume <15 cm^3^ and performed a sub-analysis in the remaining patients affected with higher tumor burden. 

### 2.2. Statistical Analysis

Statistical analyses were performed using GraphPad Prism Version 9.3.1 (GraphPad Prism Software, La Jolla, CA, USA). Mean ± standard deviation is reported for normally distributed variables as determined by the Shapiro–Wilk test. Correlations between normal organ radiotracer uptake and tumor burden were identified by using Pearson correlation coefficient or Spearman’s rank correlation coefficient (ρ). *p*-values < 0.05 were considered statistically significant.

## 3. Results

### 3.1. Patient Cohort

A total of 34 patients (mean age 62.3 ± 12.5 years; range 38–91 years; 16 females) who underwent [^68^Ga]Ga-FAPI-04 PET/CT were analyzed. Four patients received a follow-up scan, one patient underwent two follow-up scans, and thus, 40 scans were available for analysis. Twenty-four out of forty scans (60%) were performed in patients with neuroendocrine neoplasms, six (15%) with pancreatic duct adenocarcinoma, five (12.5%) with hepatocellular carcinoma, and five (12.5%) in patients with other malignancies, including colon carcinoma, sarcoma, adrenocortical carcinoma, gastrointestinal stroma tumor, and solitary fibrous tumor, respectively. Fourteen out of forty scans (35%) were carried out for primary staging, and twenty-six (55.5%) for restaging purposes. In 29 out of 40 scans (72.5%), patients had received at least one prior treatment, including surgery in 22/29 (75.9%), chemotherapy in 20/29 (69%), and external beam radiation therapy in 6/29 (20.7%) (Table 1).

### 3.2. Quantitative Assessment of Normal Organ Radiotracer Uptake and Tumor Burden

One patient had undergone prior nephrectomy on the left side due to hydronephrosis, and one patient had undergone prior splenectomy. In two patients with pancreatic duct adenocarcinoma, no normal pancreatic parenchyma could be differentiated from tumor. Moreover, in one patient with neuroendocrine tumor, normal pancreatic uptake could not be identified due to increased uptake in peripancreatic lymph node metastases. In one patient who received two scans, normal bone marrow uptake could not be determined due to diffuse metastatic involvement. As such, 1 + 1 + 2 + 1 + (2 × 3) = 11 normal-organ VOIs could not be placed, and those organs were excluded from further analysis (leaving a total of 389 normal-organ VOIs for further analyses). Highest median SUV_mean_ was recorded in the pancreas (2.15; range, 1.05–9.91), followed by the kidneys (right: 1.42; range, 0.57–3.06; left: 1.41; range, 0.73–2.97), heart (1.2; range, 0.46–2.59), spleen (0.86; range, 0.55–1.58), liver (0.65; range, 0.31–2.11), and bone marrow (0.57; range, 0.26–0.94). For tumor assessment, a total of 174 VOIs were placed (median, 3). Tumor VOIs were most often located in the liver (49/174; 28.2%), followed by lymph nodes (44/174; 25.3%), skeleton (26/174; 14.9%), the primary tumor (24/174; 13.8%), soft tissue (23/174; 13.2%), and lung (8/174; 4.6%). Descriptive statistics of uptake in normal organs and tumor lesions can be found in Table 2.

### 3.3. Correlative Assessment of Normal Organ Radiotracer Uptake and Tumor Burden

We observed a trend towards significance for uptake in the myocardium and tumor-derived SUV_max_ (ρ = 0.29, *p* = 0.07; Figure 1a) and TV (ρ = −0.30, *p* = 0.06; Figure 2a). For all other organs, no significant correlation was achieved: SUV_max_ (ρ ≤ 0.1, *p* ≥ 0.42), TV (ρ ≤ 0.11, *p* ≥ 0.43), and FTA (ρ ≤ 0.14, *p* ≥ 0.38; Table 3). Figure 1, Figure 2 and Figure 3 demonstrate correlative plots for normal organ radiotracer uptake and different parameters of tumor burden. Visual analysis showed that with increasing tumor burden, there is no obvious decrease in normal organ radiotracer uptake, as illustrated in Figure 4 in patients with respective low, intermediate, and high tumor burden.

### 3.4. Quantitative and Correlative Assessment of Normal Organ Radiotracer Uptake and Uptake in Post-Surgical Scarring

In 15/22 (68.2%) patients who received surgery before FAPI PET, visually increased tracer uptake in surgical routes or post-surgical scarring could be identified (Appendix A). A total of *n* = 31 VOIs (median, 2) were placed. Descriptive statistics of uptake in surgical routes or post-surgical scarring are listed in Appendix A.

Similar to the trend we observed in tumor-derived SUV_max_, SUV_max_ in post-surgical scarring also showed a trend towards a positive correlation with myocardial tracer uptake (Pearson’s r = 0.49, *p* = 0.06) Otherwise, there were no significant correlations between radiotracer uptake in normal organs and post-surgical scarring (Appendix A).

### 3.5. Quantitative and Correlative Assessment of Normal Organ Radiotracer Uptake and Tumor Burden in High-Tumor Burden Patients

By removing all patients with low tumor burden, the remaining 25 patients had a median sum of total tumor volume of 129.3 cm^3^. Further descriptive statistics can be found in Appendix A. 

Correlative assessment revealed a significant correlation of myocardial radiotracer uptake with tumor SUV_max_ (ρ = 0.44; *p* = 0.03), similar to the trend we observed in the entire cohort. In addition, in the sub-cohort of patients affected with high tumor burden, tumor derived FTA showed a significant correlation with liver uptake (ρ = 0.47; *p* = 0.02; Appendix A). 

## 4. Discussion

This proof-of-concept study investigated the biodistribution of [^68^Ga]Ga-FAPI-04 in normal organs as a function of total tumor burden. Except for a trend towards significance in the myocardium, which has reached significance in a sub-analysis only including high-tumor burden patients, we did not observe any relevant associations between PET-based tumor burden and normal organs. In another sub-analysis, we found no significant correlations between radiotracer uptake in normal organs and post-surgical scarring. Those preliminary findings may trigger future studies to determine possible implications for “hot” theranostic approaches and “cold” FAP-directed drugs, as one may expect an unchanged dose for normal organs even in patients with higher tumor load. 

Due to their potential as pan-tumor imaging and treatment agents, FAP-targeted radiopharmaceuticals have gained interest in recent years. Various studies have reported on excellent read-out capabilities characterized by high uptake in disease sites along with substantial low radiotracer accumulation in background tissues. Those favorable imaging results of FAPI-directed non-invasive assessment of tumor burden included, but were not limited to individuals affected with head and neck cancer, pancreas carcinoma, neuroendocrine neoplasms, gynecological tumors, or sarcoma [23,24,27,28,29,30]. Of note, this radiotracer even outperformed the most widely used PET agent in oncology, ^18^F-2-deoxy-2-fluoro-D-glucose [25], in selected clinical scenarios. Relative to the latter agent, [^68^Ga]Ga-FAPI-04 can provide a rationale to identify patients that can be scheduled for FAP-directed “hot” treatments using structurally related ß-emitting compounds [11,31]. Such an image-piloted treatment may also be applicable to patients scheduled for “cold” FAP-targeted anti-cancer drugs. In this regard, the expression of cancer-associated fibroblasts could be determined, thereby identifying individuals that would benefit the most from those medications. Such radioactive or non-radioactive therapies, however, may also be associated with relevant side effects, e.g., in the bone marrow causing substantial myelosuppression as observed for [^177^Lu]-FAP-2286 [31]. Pretherapeutic PET may then be also useful to determine relevant off-target effects prior to treatment on-set. Thus, we investigated whether normal organ uptake is associated with PET-avid tumor load, as a substantial correlation would imply potential effects in non-affected “organs at risk”, e.g., the bone marrow. First, we did not observe high median SUV_mean_ values in normal organs, ranging from only 0.57–2.15. Second, the derived correlations did not reach significance, except for the myocardium, including a negative relationship between TV and cardiac radiotracer accumulation, thereby indicating that in patients with higher tumor burden, the uptake in the heart even decreases (Figure 2a). TV, however, may not always be linear to intensity of uptake in the tumor and thus, we observed a positive, but only modest correlation between the SUV_max_ and uptake in the heart (Figure 1a), which has then reached significance in a sub-analysis only including high-tumor burden patients. Of note, this did not apply to FTA. This metric, however, may be of importance in the context of investigating a tumor-sink effect, as it analyses the activity concentration relative to the extent of volume. Thus, it allows to address a potential bias in patients with a very high uptake in one single lesion, as in those individuals, a tumor sink effect indicated by intensity alone would be negligible due to the low tumor burden. Of note, post-surgical scarring also showed no significant correlations with radiotracer uptake in normal organs, except for a trend towards a positive correlation of myocardial tracer uptake with SUV_max_ in post-surgical scar tissue. In addition, further research is warranted to tie the positive correlation of FAP intensity in the heart and tumor burden with a clinical meaning. For instance, this positive association may be relevant in tumor patients under systemic anti-tumor therapies associated with relevant cardiotoxicity [32], e.g., to identify patients prone to later cardiovascular events. As such, the herein derived findings should trigger studies to investigate FAPI-directed molecular imaging in a cardio-oncology setting beyond assessment of tumor uptake to address relevant systemic networking in high-risk individuals [33,34]. 

Taken together, our preliminary results support the notion that in patients treated with FAP-directed “cold” and “hot” therapies, the doses in the majority of normal organs would remain similar, regardless of tumor load or intensity of uptake. Despite such therapeutic implications, the observed increased FAPI uptake in the heart is in line with a recent observation by Heckmann and coworkers, which investigated oncology patients and myocardial radiotracer accumulation, reporting on focal cardiac uptake patterns in particular in patients with concomitant diagnosis of cardiovascular diseases [35]. Therefore, given our findings and the results provided by Heckmann et al. [35], future studies may also investigate systemic tumor–heart associations. Such a PET-based systemic read-out would then allow to correlate cardiac uptake and/or cardiac-tumor interrelations to clinical endpoints, thereby allowing to identify patients that may be prone to (anti-cancer therapy-related) major cardiovascular events [33,36]. 

Our preliminary results have to be interpreted with caution and should be repeated in a larger number of individuals, e.g., in a large registry study including multiple centers and a broader spectrum of tumor entities. Moreover, our subgroup analyses of patients who underwent previous surgery and patients with higher tumor burden should be substantiated in lager patient samples. In addition, the herein provided findings may not be applicable to other FAP-directed PET agents, which possess different chemical substructures [11,31,37,38]. In this regard, our study design may provide a template to assess potential tumor–organ interactions for those FAP-directed molecular imaging or therapeutic probes. Those studies may be of interest, as we and others have already observed different findings in the assessment of a potential tumor sink effect among a broad spectrum of image biomarkers used for theranostic approaches. In addition, the herein investigated radiotracer is labeled with ^68^Ga and previous studies have also reported on increased sensitivity and higher tumor detection rate if ^18^F-labeled radiotracers are used. Moreover, this has been reported in the context of theranostic PET agents for neuroendocrine tumors and prostate cancer [39,40]; such radiopharmaceutical developments and further clinical use can be anticipated in the near future for FAPI as well [41]. The resulting higher amounts of FAPI-avid tumor lesions may then trigger a re-analysis using those ^18^F-labeled FAPI PET radiotracers and may then provide more substantial correlations between tumor burden and normal organ uptake. Moreover, we report on PET-based findings which may not be necessarily applicable to potential tumor–organ interactions in patients after FAPI-directed endoradiotherapy. In this regard, comparisons derived from peri-therapeutic dosimetry studies are needed [42], e.g., by investigating tumor–organ interactions based on SPECT scans and whole-body scintigrams conducted after administration of the therapeutic radiotracer (^177^Lu-FAPI) [43]. Such an approach would then provide a more precise reflection of a potential tumor-sink effect in therapeutic setting.

## 5. Conclusions

In this proof-of-concept study, [^68^Ga]Ga-FAPI-04 PET/CT showed no relevant correlation between normal organ radiotracer uptake and tumor burden, except for a trend in the myocardium. Those preliminary findings may trigger future studies in the context of radioactive FAP-directed therapies, as higher tumor load or uptake may not lead to decreased doses in the majority of normal organs.

## Figures and Tables

**Figure 1 cancers-14-02609-f001:**
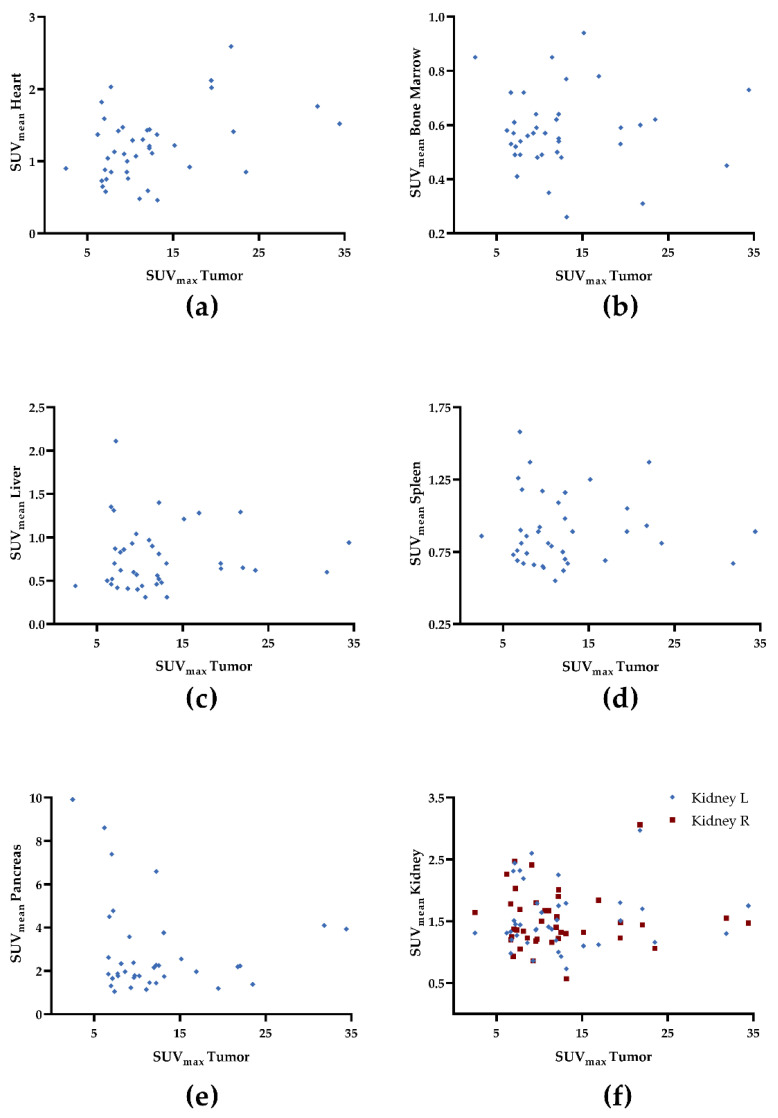
Correlative plots of tumor-derived maximum standardized uptake values (SUV_max_) and mean standardized uptake values (SUV_mean_) from organs: (**a**) heart; (**b**) bone marrow, (**c**) liver, (**d**) spleen, (**e**) pancreas, and (**f**) kidney. Rhombuses and squares are partially overlaid. No significance was reached. R = right. L = left.

**Figure 2 cancers-14-02609-f002:**
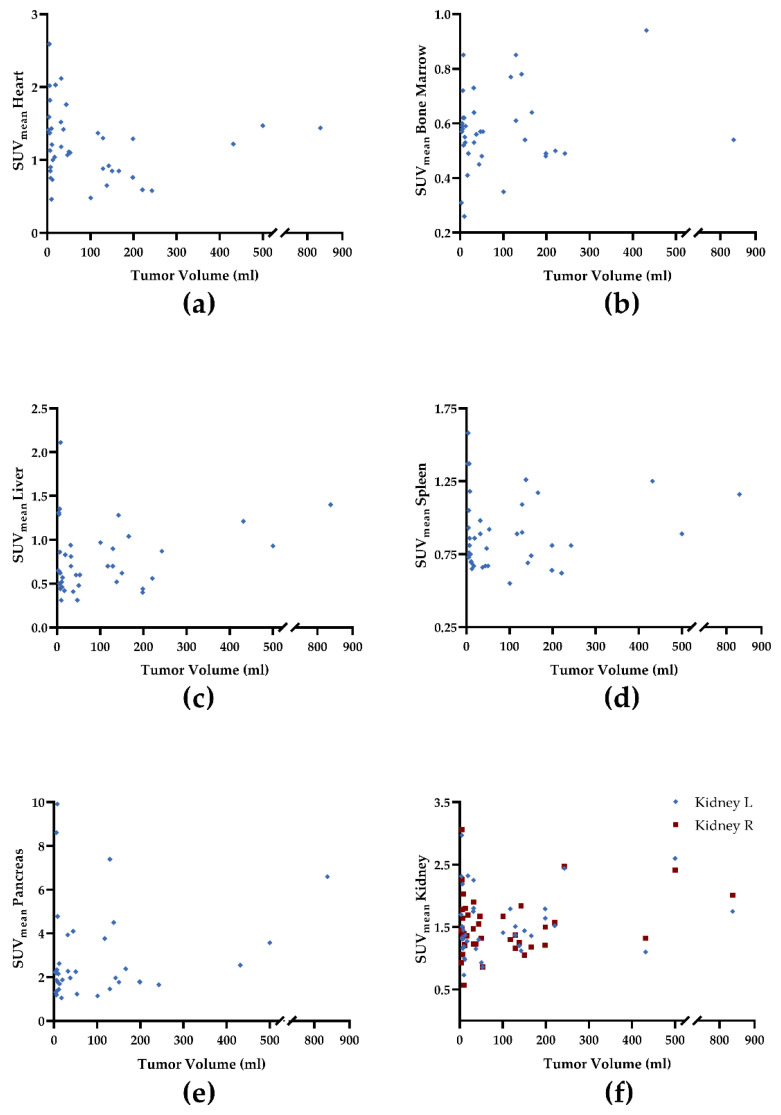
Correlative plots of PET-based tumor volume (cm^3^) and mean standardized uptake values (SUV_mean_) from organs: (**a**) heart; (**b**) bone marrow, (**c**) liver, (**d**) spleen, (**e**) pancreas, and (**f**) kidney. Rhombuses and squares are partially overlaid. No significance was reached. R = right. L = left.

**Figure 3 cancers-14-02609-f003:**
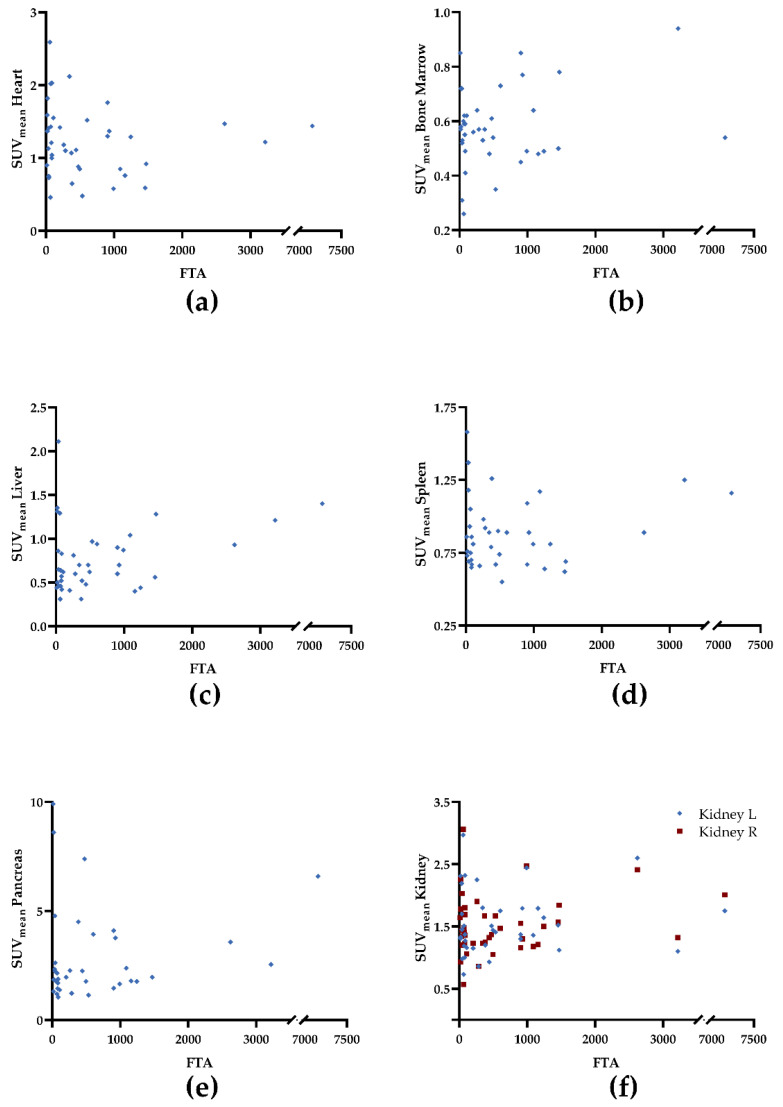
Correlative plots of fractional tumor activity (FTA) and mean standardized uptake values (SUV_mean_) from organs: (**a**) heart; (**b**) bone marrow, (**c**) liver, (**d**) spleen, (**e**) pancreas, and (**f**) kidney. Rhombuses and squares are partially overlaid. No significance was reached. R = right. L = left.

**Figure 4 cancers-14-02609-f004:**
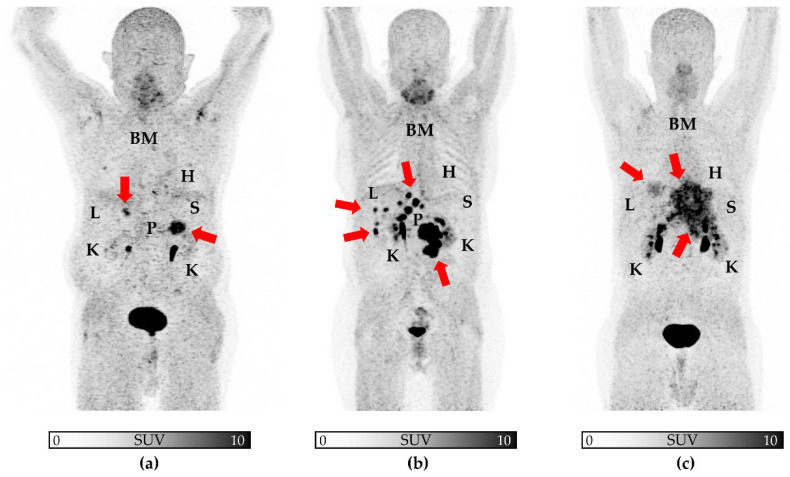
[^68^Ga]Ga-FAPI-PETs in patients with low (**a**), intermediate (**b**), and high (**c**) tumor burden. Maximum intensity projections are shown, with arrows indicating tumor burden. Visually, normal organ uptake remain similar from (**a**–**c**), supporting the notion that in patients with high tumor burden, uptake in normal organs does not drop. Given widespread disease in (**c**), the pancreas was not labelled. H = heart, L = liver, K = kidney, S = spleen, BM = bone marrow, P = pancreas.

**Table 1 cancers-14-02609-t001:** Patient characteristics.

Number of patients		34
Number of scans		40
Age ^1^		62.3 ± 12.5 years
Gender (female)		16/34 (47.1%)
Tumor entity ^2^	NEN	24/40 (60.0%)
	PDAC	6/40 (15.0%)
	HCC	5/40 (12.5%)
	Other *	5/40 (12.5%)
Indication for scan ^2^	Primary staging	14/40 (35.0%)
	Restaging	26/40 (55.0%)
Prior therapies ^2^	In total	29/40 (72.5%)
	Surgery	22/29 (75.9%)
	Chemotherapy	20/29 (69.0%)
	Radiotherapy	6/29 (20.7%)

^1^ Values are mean +/− standard deviation; ^2^ Values are number of scans, with percentages in parenthesis; NEN = neuroendocrine neoplasm, PDAC = pancreatic duct adenocarcinoma, HCC = hepatocellular carcinoma. * including sarcoma (*n* = 1), adrenocortical carcinoma (*n* = 1), colon adenocarcinoma (*n* = 1), solitary fibrous tumor (*n* = 1), and gastrointestinal stroma tumor (*n* = 1).

**Table 2 cancers-14-02609-t002:** Descriptive statistics of uptake in normal organs and tumor lesions.

		Parameter	Minimum	Median	Maximum	Mean ^1^	SD ^1^
**Normal Organs**	Heart	SUV_mean_	0.46	1.20	2.59	1.22	0.48
	BM	SUV_mean_	0.26	0.57	0.94	0.58	0.14
	Liver	SUV_mean_	0.31	0.65	2.11		
	Spleen	SUV_mean_	0.55	0.86	1.58		
	Pancreas	SUV_mean_	1.05	2.15	9.91		
	Kidney R	SUV_mean_	0.57	1.42	3.06		
	Kidney L	SUV_mean_	0.73	1.41	2.97		
**Tumor Burden**		SUV_max_	2.48	10.5	34.4		
		TV	3.00	35.0	838		
		FTA	11.8	313	7155		

^1^ mean and standard deviation (SD) are only shown for normally distributed data. BM = bone marrow, R = right, L = left, SUV_mean_ = mean standardized uptake value, SUV_max_ = maximum standardized uptake value, TV = tumor volume (in cm^3^), FTA = fractional tumor activity, defined as mean standardized uptake value × TV).

**Table 3 cancers-14-02609-t003:** Correlation (Spearman’s Rho, ρ) to determine associations between radiotracer uptake in normal organs and tumor lesions.

			Tumor Burden		
			SUV_max_	TV	FTA
**Normal Organs**	Heart	ρ	0.29	−0.30	−0.16
		*p*	0.07	0.06	0.31
	Bone Marrow	ρ	0.00	−0.05	−0.02
		*p*	0.99	0.75	0.92
	Liver	ρ	0.10	0.06	0.14
		*p*	0.53	0.73	0.38
	Spleen	ρ	0.02	−0.13	−0.13
		*p*	0.91	0.43	0.42
	Pancreas	ρ	−0.14	0.11	0.05
		*p*	0.42	0.52	0.78
	Right Kidney	ρ	−0.05	0.01	0.04
		*p*	0.75	0.97	0.80
	Left Kidney	ρ	0.00	0.06	0.10
		*p*	0.99	0.73	0.53

SUV_max_ = maximum standardized uptake value, TV = tumor volume. FTA = fractional tumor activity.

## Data Availability

The main data presented in this study are shown in the article and are available upon reasonable request from the corresponding author.

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
