# Peer review of "Associations between Normal Organs and Tumor Burden in Patients Imaged with Fibroblast Activation Protein Inhibitor-Directed Positron Emission Tomography"

_cancers, 2022, doi:10.3390/cancers14112609_

Round 1

Reviewer 1 Report

In the study, Lapa et al. analyzed in 34 patients the correlation between the uptake of [68Ga]Ga-FAPI-04 in the tumor and that in other normal organs/tissues. A trend towards significance for uptake in the myocardium and tumor-derived SUVmax was calculated. The rational for designing such a study is negotiable.

  1. Page 2 line 60: The number in the radionuclide should be superscripted.
  2. 22/29 (75.9%) of the included patients received surgery before [68Ga]Ga-FAPI-04 PET/CT scans, the authors should describe if there were any uptake in the surgical routes. From my clinical experience, massive uptake of 68Ga]Ga-FAPI-04 in the surgical areas is frequently observed. In this regard, representative images should be provided. Similarly, the uptake mediated by pancreatitis should also be described. What are the correlations between the uptake in the tumors and in the pancreatitis (or surgical areas)?
  3. What is rational accounting for the potential correlation between the uptake in the myocardium and in the tumors? The authors should discuss the potential factors or reasons underpinning the phenomenon.
  4. Although [68Ga]Ga-FAPI-04 is the tracer of choice for imaging FAP expression in clinical practice, but the fact is that other inhibitors or peptides are used for FAP-targeted therapies. So maybe there will be some difference in terms of biodistribution of different tracers/agents. In other words, the limitation should be discussed.
  5. Dosimetry results and comparisons will be more appealing and convincing.

Author Response

Review Report (Round 1)

We thank the Reviewers for their careful analysis of our manuscript and helpful comments. We greatly appreciate the opportunity to clarify our objectives and results.

Response to Reviewer 1

(...)

1. Page 2 line 60: The number in the radionuclide should be superscripted.

Ans.: Thank you for this comment. The number in the radionuclide is now superscripted.

2. A) 22/29 (75.9%) of the included patients received surgery before [68Ga]Ga-FAPI-04 PET/CT scans, the authors should describe if there were any uptake in the surgical routes. From my clinical experience, massive uptake of 68[Ga]Ga-FAPI-04 in the surgical areas is frequently observed. In this regard, representative images should be provided.

Ans.: We agree that postoperative changes in surgical routes often present with increased FAPI-uptake. After reviewing our cohort again, in 15 of 22 patients who received surgery before FAPI-imaging, visually increased tracer-uptake in surgical routes could be identified. We recorded averaged SUVmax (averaged over all areas showing increased post-surgical uptake per scan), sum of post-surgical FAP-uptake volume (PS-FV in cm³ per scan) and sum of post-surgical fractional FAP-activity (PS-FFA = PS-FV * SUVmean; per scan). Thus, a total of n = 31 VOIs were drawn. Descriptive statistics of uptake in surgical routes or post-surgical scarring are shown in Table S1.

Table S1: Descriptive statistics of uptake in surgical routes or post-surgical scarring.

Parameter

Minimum

Median

Maximum

Mean1

SD1

Post-surgical uptake

SUVmax

5.19

9.92

16.0

9.80

3.29

PS-FV

6.80

37.0

111

47.3

31.2

PS-FFA

23.4

168

767

1 mean and standard deviation (SD) are only shown for normally distributed data. SUVmax = maximum standardized uptake value, PS-FV = post-surgical FAP uptake volume, PS-FFA = post-surgical fractional FAP activity, defined as mean standardized uptake value * PS-FV).

There were no significant correlations between radiotracer uptake in normal organs and post-surgical scarring (Table S2). However, similar to the observed trend for tumor-organ associations, SUVmax in post-surgical scarring also showed a trend towards a positive correlation with myocardial tracer uptake.

Table S2. Correlation (Spearman’s Rho, ρ; or Pearson’s r) to determine associations between radiotracer uptake in normal organs and post-surgical scarring.

Post-surgical scarring

SUVmax

PS-FV

PS-FFA

Normal Organs

Heart

ρ(/r*)

0.49*

-0.30*

0.23

P

0.06

0.28

0.40

Bone Marrow

ρ(/r*)

0.02*

0.35*

0.12

P

0.94

0.20

0.66

Liver

ρ

0.00

0.06

0.03

P

>0.99

0.84

0.92

Spleen

ρ

-0.46

-0.30

-0.48

P

0.10

0.29

0.09

Pancreas

ρ

-0.27

0.01

-0.24

P

0.34

0.98

0.40

Right Kidney

ρ

0.23

-0.18

-0.04

P

0.40

0.53

0.89

Left Kidney

ρ

0.29

0.03

0.24

P

0.29

0.91

0.40

* for normally distributed data, Pearson’s r is shown. SUVmax = maximum standardized uptake value, PS-FV = post-surgical FAP uptake volume, PS-FFA = post-surgical fractional FAP activity, defined as mean standardized uptake value * PS-FV.

We now feature these analyses in the respective chapters of the manuscript. To follow the reviewer’s recommendations, we have also added a new Figure (Figure S1) showing representative radiotracer uptake in a surgical route after median laparotomy.

Similarly, the uptake mediated by pancreatitis should also be described. What are the correlations between the uptake in the tumors and in the pancreatitis (or surgical areas)?

Ans.: In our cohort, there were no patients with known pancreatitis. Additionally, the 10 patients with highest pancreatic tracer uptake did not show any morphological signs of acute or severe chronic pancreatitis on concomitant conventional imaging. For surgical areas, respective correlations with normal organ uptake have been provided in Table S1, as mentioned above.

3. What is rational accounting for the potential correlation between the uptake in the myocardium and in the tumors? The authors should discuss the potential factors or reasons underpinning the phenomenon.

Ans.: We have found a modest negative relationship between TV and myocardial radiotracer uptake, and a modest positive correlation between tumor and cardiac SUVmax. Similar to our study, a recent work by Heckmann et al. reported on myocardial FAPI-accumulation in oncologic patients, finding increased FAP-expression in patients with cardiovascular risk factors, certain chemotherapies, and a history of radiation to the chest (Heckmann et al, Circulation Cardiovasc Imaging, 2020). These findings point towards systemic tumor-heart interactions, and thus, may trigger future studies to link FAP-derived radiotracer accumulation in sites of disease with cardiovascular outcome, thereby allowing to identify high-risks prone to later cardiovascular events, e.g., due to overzealous cardiac fibrotic remodeling in patients under anti-tumor therapies. In this regard, our findings shed light on potential tumor-heart interactions in a cardio-oncology setting, and thus, may lead to future studies investigating relevant myocardial off-target effects during anti-cancer treatment, e.g., cardiotoxicity under chemotherapy or radiation to the chest. We now elaborate on this in the Discussion section of the manuscript (lines 259 ff.)

4. Although [68Ga]Ga-FAPI-04 is the tracer of choice for imaging FAP expression in clinical practice, but the fact is that other inhibitors or peptides are used for FAP-targeted therapies. So maybe there will be some difference in terms of biodistribution of different tracers/agents. In other words, the limitation should be discussed.

Ans.: Thank you for pointing this out. We now address this in the limitation section: “[…] we and others have already observed different findings in the assessment of a potential tumor sink effect among a broad spectrum of image biomarkers used for theranostic approaches. In addition, the herein investigated radiotracer is labeled with 68Ga and previous studies have also reported on increased sensitivity and higher tumor detection rate if 18F-labeled radiotracers are used. Also this has been reported in the context of theranostic PET agents for neuroendocrine tumors and prostate cancer, such radiopharmaceutical developments and further clinical use can be anticipated in the near-term future for FAPI as well. The resulting higher amounts of FAPI-avid tumor lesions may then trigger a re-analysis using those 18F-labeled FAPI PET radiotracers and may then provide more substantial correlations between tumor burden and normal organ uptake.” (lines 290 ff.)

5. Dosimetry results and comparisons will be more appealing and convincing.

Ans.: FAPI-directed PET/CT may also serve a non-invasive read-out for the target expression in the context of non-radiolabeled medication, e.g., to determine patients that are eligible for treatment with “cold” FAPI-directed drugs. In such a clinical scenario, dosimetry-based results would not be available. Nonetheless, in the context of “hot” theranostic approaches, we absolutely agree with the reviewer that dosimetry results derived from posttherapeutic SPECTs or whole-body planar scintigrams would provide a more precise reflection of a potential tumor-sink effect. Those aspects have been added on p. 11 (lines 302ff).

Reviewer 2 Report

This manuscript investigates if there is a correlation in the [68Ga]Ga-FAPI-04 radiotracer uptake in normal organs and extent of tumor burden. Their preliminary findings based on a small cohort indicates that there is no significant correlation between normal uptake and tumor burden except for a mild correlation with uptake in heart. I think the manuscript is well organized and clearly written, and aims to address an important question in terms of different FAPI related therapies (drugs, therapy radio-ligands). Despite the limited amount of studies analyses based on SUVmax, SUVmean, tumor volume, and fractional tumor activity (FTA) appear to show consistent results.

However, I do have an important concern with the distribution of the data as low-tumor burden samples are more populated than higher-tumor burden ones. There is a large variation in normal uptake when the tumor burden is low, but we cannot see this for the higher-tumor region as there is not much datapoints. Part of this variation might be related to the PVE although it is stated that only >15mm lesions were considered. It seems to me that if these low-tumor burden datapoints (about 10-15 of them) excluded some correlation might be found although then the analyses would be limited to an even smaller sample.

Could you discuss what are the possible reasons for not finding such a correlation. From a physical standpoint only, without considering the biology, I would expect higher uptake in tumors implies less uptake in normal tissue for a given activity in body.

Few minor edits:

L100 “averaged maximum standardized uptake values”. As I understand averaged over all the tumors, but please clarify this in the text.

L102 “To  avoid partial volume effects, lesions smaller than 15 mm or 1.7 cm3 were not included” > maybe say “reduce” instead of avoid, since with the given PET resolution there will be still some quantification issues of PVE although I doubt these inaccuracies would have any considerably affects in the results.

Table2: Para-meter -> Parameter

Table2 –Please add the unit for TV in footnote: TV = tumor volume (cm3?).

Author Response

Response to Reviewer 2

(…)

However, I do have an important concern with the distribution of the data as low-tumor burden samples are more populated than higher-tumor burden ones. There is a large variation in normal uptake when the tumor burden is low, but we cannot see this for the higher-tumor region as there is not much datapoints. Part of this variation might be related to the PVE although it is stated that only >15mm lesions were considered. It seems to me that if these low-tumor burden datapoints (about 10-15 of them) excluded some correlation might be found although then the analyses would be limited to an even smaller sample. Could you discuss what are the possible reasons for not finding such a correlation. From a physical standpoint only, without considering the biology, I would expect higher uptake in tumors implies less uptake in normal tissue for a given activity in body.

Ans.: Thank you for pointing this out. Following the reviewer’s recommendation, we removed all low-tumor burden scans with a sum of total tumor volume < 15 cm3 (n = 15). The remaining sample of only n = 25 scans had a median sum of total tumor volume of 129.3 cm3 and now showed a significant correlation of myocardial radiotracer uptake with tumor SUVmax (ρ = 0.44; p = 0.03), similar to the trend we observed in the original patient cohort (ρ = 0.29; p = 0.07). In addition, after removing low-tumor burden patients, tumor derived FTA showed a significant correlation with liver uptake (ρ = 0.47; p = 0.02), which further demonstrates that exclusion of low tumor-burden datapoints may have a significant impact on the derived correlations. Those analyses, however, should be still interpreted with caution, as they were obtained from a limited sample size.

To grant Reviewer 2 an improved overview of the presented data, we provide Tables S3 and S4 for the above mentioned high-tumor burden sample:

Table S3 (‘high-tumor burden’ only, n = 25): Descriptive statistics of uptake in normal organs and tumor lesions.

Parameter

Minimum

Median

Maximum

Mean1

SD1

Normal Organs

Heart

SUVmean

0.48

1.11

2.12

1.16

0.43

BM

SUVmean

0.35

0.54

0.94

0.58

0.15

Liver

SUVmean

0.31

0.70

1.40

0.75

0.29

Spleen

SUVmean

0.55

0.86

1.26

0.86

0.20

Pancreas

SUVmean

1.05

2.11

7.39

Kidney R

SUVmean

0.86

1.37

2.47

1.50

0.39

Kidney L

SUVmean

0.86

1.48

2.60

1.57

0.47

Tumor Burden

SUVmax

6.77

10.7

34.4

TV

17.4

129.3

838

FTA

84.3

603.9

7155

1 mean and standard deviation (SD) are only shown for normally distributed data. BM = bone marrow, R = right, L = left, SUVmean = mean standardized uptake value, SUVmax = maximum standardized uptake value, TV = tumor volume, FTA = fractional tumor activity, defined as mean standardized uptake value * TV).

Table S4 (‘high tumor burden’ only, n = 25). Correlation (Spearman’s Rho, ρ) to determine associations between radiotracer uptake in normal organs and tumor lesions.

‘High’ Tumor Burden

SUVmax

TV

FTA

Normal Organs

Heart

ρ

0.44

-0.34

-0.11

P

0.03

0.09

0.60

Bone Marrow

ρ

0.27

0.16

0.22

P

0.21

0.48

0.32

Liver

ρ

0.31

0.31

0.47

P

0.13

0.13

0.02

Spleen

ρ

-0.03

0.19

0.10

P

0.87

0.35

0.64

Pancreas

ρ

0.23

0.16

0.27

P

0.30

0.47

0.23

Right Kidney

ρ

0.09

0.13

0.23

P

0.67

0.53

0.22

Left Kidney

ρ

-0.06

0.11

0.12

P

0.78

0.60

0.57

SUVmax = maximum standardized uptake value, TV = tumor volume. FTA = fractional tumor activity.

We now feature these analyses in the respective chapters of the manuscript on p. 3, l.112 ff.; p. 5, l. 188 ff; p. 9 l 220 f; p. 10, l. 253 f).

Few minor edits:

  • L100 “averaged maximum standardized uptake values”. As I understand averaged over all the tumors, but please clarify this in the text.

Ans.: Thank you for this comment. For every scan, SUVmax values were averaged over all the tumor lesions found. We now clarify this in the manuscript on p. 3, l. 103 f.

  • L102 “To avoid partial volume effects, lesions smaller than 15 mm or 1.7 cm3 were not included” > maybe say “reduce” instead of avoid, since with the given PET resolution there will be still some quantification issues of PVE although I doubt these inaccuracies would have any considerably affects in the results.

Ans.: We thank the reviewer for raising this issue to our attention. We have corrected the manuscript accordingly (p. 3, l. 109).

  • Table2: Para-meter -> Parameter

Ans.: We have corrected the manuscript accordingly.

  • Table2 –Please add the unit for TV in footnote:TV = tumor volume (cm3?).

Ans.: We have corrected the manuscript accordingly.

Reviewer 3 Report

This is a single-center retrospective study with contained 34 patients with different malignant tumors, which aimed to quantitatively investigate 68Ga-FAPI-04 uptake in normal organs and to assess the relationship with the extent of FAPI-avid tumor burden. It's an interesting idea, but the sample size is limited, the data and statistical analysis are slightly simple, so a negative conclusion is not unexpected ,and it has little significance for clinical practice. Therefore, it’s recommended for rejection.

Author Response

Response to Reviewer 3

This is a single-center retrospective study with contained 34 patients with different malignant tumors, which aimed to quantitatively investigate 68Ga-FAPI-04 uptake in normal organs and to assess the relationship with the extent of FAPI-avid tumor burden. It's an interesting idea, but the sample size is limited, the data and statistical analysis are slightly simple, so a negative conclusion is not unexpected, and it has little significance for clinical practice.

Ans.: Indeed, these are important aspects. The herein used statistical analyses in the context of assessing a potential tumor sink effect in patients imaged with theranostic PET agents has already been used previously, e.g., by Gafita et al (J Nucl Med, 2022), Serfling et al (Mol Imag Biol, 2022) or Beauregard et al (EJNMMI, 2012). Although we have enrolled a limited sample size, we still think that our findings may trigger future research, preferably in a larger setting. In addition, although the number of investigated patients was rather low, we still observed significant correlations, in particular for the myocardium among the entire cohort. Following the recommendation of the second reviewer, our findings have been further corroborated in a subanalysis investigating an even lower number of subjects (exclusively having high tumor burden; novel Supplementary Tables S3 and S4). Nonetheless, we totally agree with the reviewer that more substantial conclusions can be drawn if more patients are enrolled, preferably in a multicenter setting. In this regard, our feasibility study may provide a template for such a study design enrolling more subjects imaged with FAPI-directed PET. This has been addressed on p. 10, l. 283 ff.

Round 2

Reviewer 1 Report

The authors have adequately answered my comments.

Author Response

The authors have adequately answered my comments.

Ans.: We thank the reviewer for this encouraging comment.

Reviewer 2 Report

The authors address my main concern with their sub-analysis. They should also mention these findings in the abstract. Otherwise it looks inconsistent with section 3.5.

L111-113 "Since....levels." is not a complete sentence. So this paragraph should be revised: Since the potential impact of tumor burden on normal organ radiotracer uptake would be expected to be more pronounced at higher tumor burden levels. Thus, by removing 113 patients with a sum of total tumor volume < 15 cm3, we also performed an additional subgroup analysis in a cohort with higher tumor burden

circular volumes -> spherical volumes? 

Author Response

Response to Reviewer 2

(...)

The authors address my main concern with their sub-analysis. They should also mention these findings in the abstract. Otherwise it looks inconsistent with section 3.5.

Ans.: This has been addressed. The abstract now reads: “In a sub-analysis exclusively investigating patients with high tumor burden, significant correlations of myocardial uptake with tumor SUVmax (ρ = 0.44; p = 0.03) and tumor-derived FTA with liver uptake (ρ = 0.47; p = 0.02) were recorded.”.

L111-113 "Since....levels." is not a complete sentence. So this paragraph should be revised: Since the potential impact of tumor burden on normal organ radiotracer uptake would be expected to be more pronounced at higher tumor burden levels. Thus, by removing 113 patients with a sum of total tumor volume < 15 cm3, we also performed an additional subgroup analysis in a cohort with higher tumor burden. 

Ans.: We thank the reviewer for bringing this issue to our attention. This has been corrected

on l.113: “The potential impact of tumor burden on normal organ radiotracer uptake would be

expected to be more pronounced at higher tumor burden levels. Thus, we removed patients with a sum of total tumor volume < 15 cm3 and performed a sub-analysis in the remaining patients affected with higher tumor burden.”

circular volumes -> spherical volumes? 

Ans.: This has been corrected on l. 103.

Reviewer 3 Report

Please kindly check the manuscript to aviod the mistakes.

Author Response

Response to Reviewer 3

Please kindly check the manuscript to aviod the mistakes.

Ans.: The manuscript has been checked again for mistakes and has been modified accordingly

(changes highlighted in red).
